# Free-space optical spiking neural network

**Reyhane Ahmadi[1], Amirreza Ahmadnejad[2], Somayyeh Koohi[1] ***

**1** Department of Computer Engineering, Sharif University of Technology, Tehran, Iran, **2** Department of Electrical Engineering, Sharif University of Technology, Tehran, Iran

\* koohi@sharif.edu

## Abstract

Neuromorphic engineering has emerged as a promising avenue for developing brain-inspired computational systems. However, conventional electronic AI-based processors often encounter challenges related to processing speed and thermal dissipation. As an alternative, optical implementations of such processors have been proposed, capitalizing on the intrinsic information-processing capabilities of light. Among the various Optical Neural Networks (ONNs) explored within the realm of optical neuromorphic engineering, Spiking Neural Networks (SNNs) have exhibited notable success in emulating the computational principles of the human brain. The event-based spiking nature of optical SNNs offers capabilities in low-power operation, speed, temporal processing, analog computing, and hardware efficiency that are difficult or impossible to match with other ONN types. In this work, we introduce the pioneering Free-space Optical Deep Spiking Convolutional Neural Network (OSCNN), a novel approach inspired by the computational model of the human eye. Our OSCNN leverages free-space optics to enhance power efficiency and processing speed while maintaining high accuracy in pattern detection. Specifically, our model employs Gabor filters in the initial layer for effective feature extraction, and utilizes optical components such as Intensity-to-Delay conversion and a synchronizer, designed using readily available optical components. The OSCNN was rigorously tested on benchmark datasets, including MNIST, ETH80, and Caltech, demonstrating competitive classification accuracy. Our comparative analysis reveals that the OSCNN consumes only 1.6 W of power with a processing speed of 2.44 ms, significantly outperforming conventional electronic CNNs on GPUs, which typically consume 150-300 W with processing speeds of 1-5 ms, and competing favorably with other free-space ONNs. Our contributions include addressing several key challenges in optical neural network implementation. To ensure nanometer-scale precision in component alignment, we propose advanced micro-positioning systems and active feedback control mechanisms. To enhance signal integrity, we employ high-quality optical components, error correction algorithms, adaptive optics, and noise-resistant coding schemes. The integration of optical and electronic components is optimized through the design of high-speed opto-electronic converters, custom integrated circuits, and advanced packaging techniques. Moreover, we utilize highly efficient, compact semiconductor laser diodes and develop novel cooling strategies to minimize power consumption and footprint.

**Data Availability Statement:** All relevant data for this study are publicly available from the GitHub repository (https://github.com/AAhmadnejad98/OSCNN.git). This study used the following publicly available open source third-party datasets: MNIST (https://www.kaggle.com/datasets/hojjatk/mnist-

dataset), Caltech (https://www.kaggle.com/datasets/jessicali9530/caltech256), and ETH-80 (https://github.com/chenchkx/ETH-80).

**Funding:** The author(s) received no specific funding for this work.

**Competing interests:** The authors have declared that no competing interests exist.

# 1 Introduction

The human brain represents a profoundly intricate and remarkable biological entity. The endeavor to engineer a computational processor possessing commensurate attributes in power, precision, integration, and speed has perennially constituted a paramount aspiration for processor designers. Neuromorphic Engineering stands as a foundational paradigm facilitating the realization of such processors, primarily through the incorporation of neural network architectures [1–3]. Despite the notable achievements resulting from this approach [4–7], the central challenge in processor design endures as the demand for processing voluminous datasets continues to burgeon. One of the fundamental problems that has led researchers to the optical implementation of processing and even neuromorphic structures in recent years is related to Moore's law [7].

To address this persistent challenge, Optical Neuromorphic Engineering has emerged as a novel and innovative domain [8]. Optical Neuromorphic Engineering exploits light's distinctive attributes, including its exceptional propagation speed and the extended degrees of freedom it affords in comparison to electrons, encompassing characteristics such as frequency, phase, polarization, and mode. Furthermore, optical systems manifest reduced loss, rendering them remarkably compelling for the design and construction of Optical Neural Networks (ONNs). The most significant advancements in recent years are documented in [9–12] and comprehensively reviewed in [13–15].

Various Optical Neural Networks (ONNs) have been explored within the realm of optical neuromorphic engineering. Among these, Spiking Neural Networks (SNNs) have exhibited notable success in emulating the computational principles of the human brain. SNNs, which constitute a class of neural networks that emulate the structural and functional aspects of the human brain, have garnered significant attention in this context. Numerous optical models have been proposed for implementing SNNs; however, these models have been primarily integrated. Several intricate designs featuring components such as Vertical-Cavity Surface-Emitting Lasers (VCSELs) [16], micro-ring resonators [17], and phase-change materials [18] have been suggested. Nonetheless, these designs prove ill-suited for the demanding task of high-volume data processing, thereby underscoring the formidable challenges encountered within Optical Neuromorphic Engineering.

The potential need for optical-spiking neural networks arises from the unique combination of optical and spiking neural network properties, offering specific advantages in certain applications. While all ONNs share standard optical features, incorporating spiking neural network characteristics introduces additional benefits, which can mention as help to have Event-Driven Processor Architectures, Temporal Information Capture, and Enhanced Robustness to Noisy Inputs.

Implementing Spiking Neural Networks (SNNs) in an Optical Fiber System (OFS) format offers several advantages compared to integrated photonic implementations:

- **Increased Scalability**: OFS systems can accommodate vast numbers of optical components, enabling the realization of networks with millions or billions of spiking neurons. This massive scalability is challenging for integrated photonics.

- **Flexible Reconfigurability**: The OFS approach simplifies reconfiguring and modifying network connectivity patterns by adjusting optical components. Integrated systems are more rigid once fabricated. Additionally, the broader operating wavelengths of OFS implementations can leverage a more comprehensive range of optical wavelengths, from UV to far infrared, whereas on-chip photonics have much narrower wavelength compatibility.

- **Simplified Training**: Specific OFS architectures allow direct intensity-to-spike training via simple optical nonlinearities, whereas integrated training often requires more complex components.

- **Lower Optical Loss**: OFS systems can reduce loss, allowing for more extensive networks. Moreover, hybrid electronic-photonic operation interfaces between OFS optical components and traditional electronic neurons/synapses provide unique hybrid SNN capabilities.

It is imperative to establish precise mathematical models for each constituent component to develop an OFS model for SNNs. Remarkably successful models rooted in neuroscience [19, 20] have been devised, emulating the structural attributes responsible for object detection within the human eye. In this paper, we aim to draw upon these well-established models as a source of inspiration for designing OFS components, thereby facilitating the simulation of the Free-Space Optical deep Spiking Convolutional Neural Network (OSCNN). To the best of our knowledge, OSCNN marks the inaugural foray into the realm of OFS modeling for SNNs, encompassing critical elements such as Gabor filters for feature extraction, intensity-to-delay conversion, synchronization mechanisms, convolution layers, max-pooling procedures, and a classification framework. A dedicated module was introduced to execute the intensity-to-delay conversion, employing a Spatial Light Modulator (SLM) after the feature extractor layer.

Moreover, an optical synchronizer has been meticulously devised to address the temporal processing aspects inherent to optic signals. Ensuring that the time order of signals remains intact post-convolution, this synchronizer draws inspiration from the Free-Space Optical delay line concept [21]. The performance of OSCNN was systematically evaluated across three distinct datasets: MNIST, Caltech, and ETH80. OSCNN demonstrated notable achievements, boasting significant performance metrics compared to electronic Neural Networks (NNs) and Optical Neural Networks (ONNs). Notably, Gabor filters were harnessed as feature extractors in the initial model layer, with evaluations conducted under both trained and fixed conditions. While alternative filters, such as Canny, Laplacian, and Sobel, were explored as feature extraction mechanisms, the most favorable outcomes for OSCNN were attained using Gabor filters. The results underscore the versatility of Gabor-form convolutional kernels, revealing their efficacy in image and time-series processing applications [22, 23].

The general structure of this article is as follows: firstly, the optical model of the OSCNN is discussed. We have designed this model based on separate components, which have been explained in detail regarding the mathematics and how to model each component. Then, the numerical simulation results are presented, including a comparison of the overall performance of integrated optical neural networks and free space in the context of data classification, as well as an analysis of the effectiveness concerning noise, power, and required speed. Finally, we discuss future works and present our conclusions.

## 2 OSCNN model

In delineating the architectural framework of the Optical Free Space Spiking Convolutional Neural Network (OSCNN), it is imperative to establish a comprehensive model for neurons within Spiking Neural Networks (SNNs, also known as integrate-and-fire models). SNNs fundamentally operate on spike-timing-dependent plasticity (STDP) principles, a paradigm necessitating optical modeling to faithfully replicate its mechanisms. OSCNN adopts a specialized variation of computational neurons, acknowledging that forthcoming research will delve into developing a more precise neuron model. As an alternative to STDP, backpropagation (BP) can be employed for training the OSCNN model, as described in [24]. Subsequently, this section elaborates on the mathematical underpinnings of each module within OSCNN, along

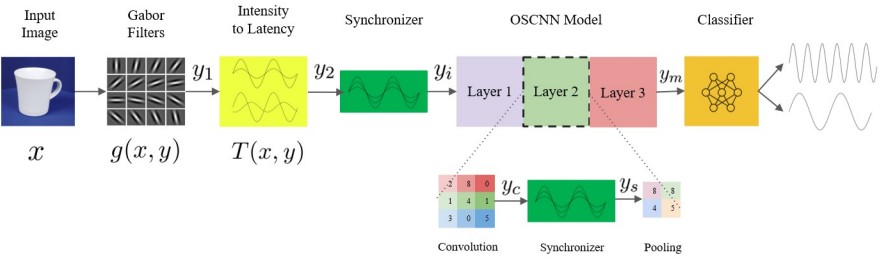

**Fig 1. Outline of OSCNN.**

with their optical equivalents. The holistic structure of the OSCNN model is illustrated in Fig 1 for a comprehensive overview.

As shown in Fig 1, which is inspired by [20], the main features are first extracted from the input data. In the case of SNNs, these features are represented as a series of lines with different thicknesses, indicating their importance. These thickness-based features are then converted into fuzzy features, where those with higher value and importance are released into the model earlier. A crucial aspect of this model is the synchronization of different phases or speeds, ensuring that all features begin propagating through the main neural network simultaneously. For this reason, a synchronizer is employed. The main OSCNN model comprises the convolution layer, synchronizer, and pooling layer. Finally, a classifier categorizes the input data.

To further explain the main idea of [20] during test, the mathematical procedure can be expanded as follows. First, they apply Difference of Gaussians (DoG) filters to the input image to detect contrasts. This can be mathematically represented as:

$$\mathrm{DoG}(x, y) = I(x, y) * (G_{\sigma_1}(x, y) - G_{\sigma_2}(x, y)) \tag{1}$$

where $I(x, y)$ is the input image, $G_\sigma(x, y)$ is a Gaussian filter with standard deviation $\sigma$, and $*$ denotes convolution. Next, the strength of these contrasts is encoded into spike times. The encoding is such that higher contrasts result in shorter latencies:

$$t_{ij} = T\left(1 - \frac{\mathrm{DoG}_{ij}}{\max(\mathrm{DoG})}\right) \tag{2}$$

where $t_{ij}$ is the spike time for the pixel at position $(i, j)$, $T$ is a constant representing the maximum possible latency, and $\mathrm{DoG}_{ij}$ is the DoG response at $(i, j)$.

The network consists of several layers of convolutional and pooling neurons. Neurons in the convolutional layers detect features by integrating input spikes:

$$V_{ij}(t) = \sum_{k,l} w_{kl} S_{i+k,j+l}(t - d_{kl}) \tag{3}$$

where $V_{ij}(t)$ is the membrane potential at neuron $(i, j)$ at time $t$, $w_{kl}$ are the synaptic weights, and $S_{i+k,j+l}(t - d_{kl})$ represents the incoming spikes from neurons at position $(i + k, j + l)$ with delay $d_{kl}$. The learning mechanism employed is Spike-Timing-Dependent Plasticity (STDP), which updates the synaptic weights based on the timing of pre- and postsynaptic spikes:

$$\Delta w = \begin{cases} A_+ e^{-\Delta t/\tau_+} & \text{if } \Delta t > 0 \\ -A_- e^{\Delta t/\tau_-} & \text{if } \Delta t \leq 0 \end{cases} \tag{4}$$

where $\Delta w$ is the change in synaptic weight, $\Delta t$ is the difference in spike timing between pre-

and postsynaptic neurons, and $A_+$, $A_-$, $\tau_+$, $\tau_-$ are constants. A Winner-Take-All mechanism ensures that only the earliest spikes are considered, suppressing later spikes:

$$V_{ij}(t) = \begin{cases} V_{ij}(t) & \text{if } t = \min(t_{ij}) \\ 0 & \text{otherwise} \end{cases} \tag{5}$$

Pooling layers provide translation invariance and compress visual information. Translation invariance is achieved using maximum operations:

$$P_{ij} = \max_{(m,n)\in\mathcal{R}(i,j)} V_{mn}(t) \tag{6}$$

where $P_{ij}$ is the pooled response at position $(i, j)$ and $\mathcal{R}(i, j)$ is the receptive field centered at $(i, j)$. The earliest spike times are propagated to reduce data complexity:

$$t_{P_{ij}} = \min_{(m,n)\in\mathcal{R}(i,j)} t_{V_{mn}} \tag{7}$$

where $t_{P_{ij}}$ is the spike time for the pooled response at $(i, j)$ and $t_{V_{mn}}$ are the spike times from the convolutional layer. The classifier interprets the activity of neurons in the final pooling layer to determine the object category:

$$\hat{y} = \arg \max_{y} f(P, W) \tag{8}$$

where $\hat{y}$ is the predicted category, $f$ is a function mapping pooled responses $P$ and classifier weights $W$ to category scores.

In the following, our goal is to simulate and present the optical equivalent of each of these functions, inspired by the mathematics and functions used to implement the [20]. The following sections provide a detailed explanation of each block and their free space optical equivalent design.

## 2.1 Implementation of Gabor filters using an optical 4f system

A Gabor filter can be implemented using an optical 4f system, which is a common setup for performing Fourier transforms and convolutions optically. The 4f system consists of two lenses with focal length $f$, placed at a distance $2f$ apart. This system utilizes the Fourier transform properties of lenses to perform the convolution of an image with a Gabor filter in the Fourier domain. The key components of the setup include the input image $X(x, y)$ placed at the input plane. A lens with focal length $f$ performs the Fourier transform of the input image, producing $X(u, v)$ in the Fourier plane. In the Fourier plane, a Spatial Light Modulator (SLM) is used to impose the Gabor filter $G(u, v)$ on the Fourier-transformed image. The second lens, also with focal length $f$, performs the inverse Fourier transform to obtain the convolved output image in the output plane. Finally, the output image $y_1(x, y)$ is obtained at this plane.

The mathematical formulation of the 4f system involves the Fourier transform by the first lens. If the input image is $X(x, y)$, then the lens transforms it to the Fourier domain:

$$X(u, v) = \mathcal{F}\{X(x, y)\} \tag{9}$$

The Fourier-transformed image is multiplied by the Gabor filter $G(u, v)$ imposed by the SLM:

$$Y(u, v) = X(u, v) \cdot G(u, v) \tag{10}$$

The product is then transformed back to the spatial domain by the second lens:

$$y_1(x, y) = \mathcal{F}^{-1}\{Y(u, v)\} \tag{11}$$

Combining these steps, the overall convolution of the input image with the Gabor filter is given by:

$$y_1(x, y) = \mathcal{F}^{-1}\{X(u, v) \cdot G(u, v)\} \tag{12}$$

## 2.2 Optical intensity to phase conversion

The conversion of intensity-modulated information into phase information is crucial for optical neural networks (ONNs), enabling efficient processing and representation of data using light. This transformation, known as intensity-to-phase (latency) conversion, is implemented through a Spatial Light Modulator (SLM) in an optical setup. The optical setup for intensity-to-phase conversion typically includes the input plane, where the intensity-modulated image $y_1(u, v)$ is located, derived from the preceding convolution module. The SLM is positioned in the Fourier plane, and it introduces phase shifts according to the intensity values of the input image. The output plane generates the phase-modulated output $y_2(u, v)$ after passing through the SLM.

The intensity-to-phase conversion process can be mathematically described as follows: Given the input image $y_1(u, v)$ with associated intensity values $I(u, v)$, the conversion operation is represented by:

$$I(u, v) = I_0 \, \cos^2(\phi(u, v)) \tag{13}$$

where $I_0$ denotes the maximum intensity and $\phi(u, v)$ represents the phase values to be achieved. To achieve this conversion optically using the SLM, we utilize a complex transmission function $T(u, v)$, where:

$$T(u, v) = e^{i\phi(u,v)} \tag{14}$$

Here, $e^{i\phi(u,v)}$ denotes the phase modulation introduced by the SLM. The output $y_2(u, v)$ from the intensity-to-phase conversion module is then given by:

$$y_2(u, v) = y_1(u, v) \cdot T(u, v) \tag{15}$$

where $y_1(u, v)$ is the intensity-modulated input image and $T(u, v)$ is the complex transmission function imposed by the SLM.

## 2.3 Optical synchronizer

Upon converting intensity to phase, ensuring synchronized emission of optical signals from a common temporal reference point becomes essential, especially in Optical Free Space (OFS) setups. Traditional delay line structures used in integrated Optical Neural Networks (ONNs) are replaced in OFS by innovative techniques like diffraction gratings and Spatial Light Modulators (SLMs). The synchronizer in OFS employs a diffraction grating and SLM based on principles outlined in the literature. Let $y_2(u, v)$ denote the output field from the intensity-to-phase conversion module. The diffraction grating has a pitch $\Lambda$ and a diffraction angle $\theta$. It disperses the input field into multiple diffraction orders, each carrying a version of the input signal delayed by $\tau_m = m\Lambda \sin \theta/c$, where $m$ is the diffraction order and $c$ is the speed of light. The electric field $E_m(u, v)$ associated with the $m$th diffraction order is thus given by:

$$E_m(u, v) = y_2(u, v) \exp\left(i\frac{2\pi m\Lambda\sin\theta}{c}\right) \tag{16}$$

After diffraction, the diffraction orders undergo phase modulation using an SLM. The phase modulation function introduced by the SLM is represented by a complex-valued function $f(u, v)$. A lens with focal length $f$ recombines the electric fields of the diffraction orders. The output signal $y_i(u, v)$ at the focal point of the lens is expressed as:

$$y_i(u, v) = \sum_{m=-\infty}^{\infty} E_m(u, v)\exp(if(u, v)) \cdot \text{sinc}\left(\frac{m\Lambda\sin\theta}{f}\right) \tag{17}$$

where $\text{sinc}(x) = \frac{\sin(\pi x)}{\pi x}$. In practical implementations, a finite number of diffraction orders, typically up to $m = 20$, are considered due to computational feasibility and system constraints.

The optical setup includes the input field $y_2(u, v)$ from the intensity-to-phase module. The diffraction grating produces multiple diffraction orders. The SLM modulates the phase of each diffraction order. A lens reconstructs the combined signal at its focal point. The output signal $y_i(u, v)$ is synchronized and ready for further processing. This optical synchronizer ensures synchronized processing of optical signals in OFS environments, enabling efficient neural network operations.

## 2.4 Layers and classifier

Following the feature extraction phase, optical signals need to be integrated to capture the essential features embedded within the image. This integration is accomplished through a 3-layer system comprising a convolution layer, a synchronizer, and a max-pooling module. The convolution layer combines optical signals using trainable kernels in a 4f correlator configuration. Simultaneously, the max-pooling module identifies critical features using another 4f correlator setup, integrating a saturable absorber (SA) for nonlinearity.

The convolution operation in optical systems is realized using a 4f correlator, where the input signal $y_2(u, v)$ undergoes convolution with trainable kernels represented by $W(u, v)$:

$$y_{\text{conv}}(u, v) = y_2(u, v) \star W(u, v) \tag{18}$$

Here, $\star$ denotes the convolution operation, and $W(u, v)$ represents the learnable weights. To maintain temporal coherence after convolution, a synchronizer using a diffraction grating and SLM adjusts the phase of each signal component. The electric field $E_m(u, v)$ associated with the $m$th diffraction order is modulated with a phase shift $f(u, v)$ and filtered through a sinc function:

$$y_{\text{sync}}(u, v) = \sum_{m=-\infty}^{\infty} E_m(u, v)\exp(if(u, v)) \cdot \text{sinc}\left(\frac{m\Lambda\sin\theta}{f}\right) \tag{19}$$

where $\text{sinc}(x) = \frac{\sin(\pi x)}{\pi x}$.

The max-pooling function preserves critical features by selecting the maximum value within a defined region, implemented similarly in a 4f correlator structure with a saturable absorber (SA):

$$y_{\text{pool}}(u, v) = \max(y_{\text{sync}}(u, v)) \tag{20}$$

The SA introduces nonlinear behavior akin to traditional neural network layers. The classifier module employs a multilayer neural network structure incorporating the saturable absorber (SA) for nonlinearity. This optical emulation of the classifier is designed to classify input data effectively.

The general scheme of OSCNN optical setup is shown in Fig 2. As represented in Fig 2, we first have a continuous wave laser at 1550 nm, which is emitted onto the first SLM to encode

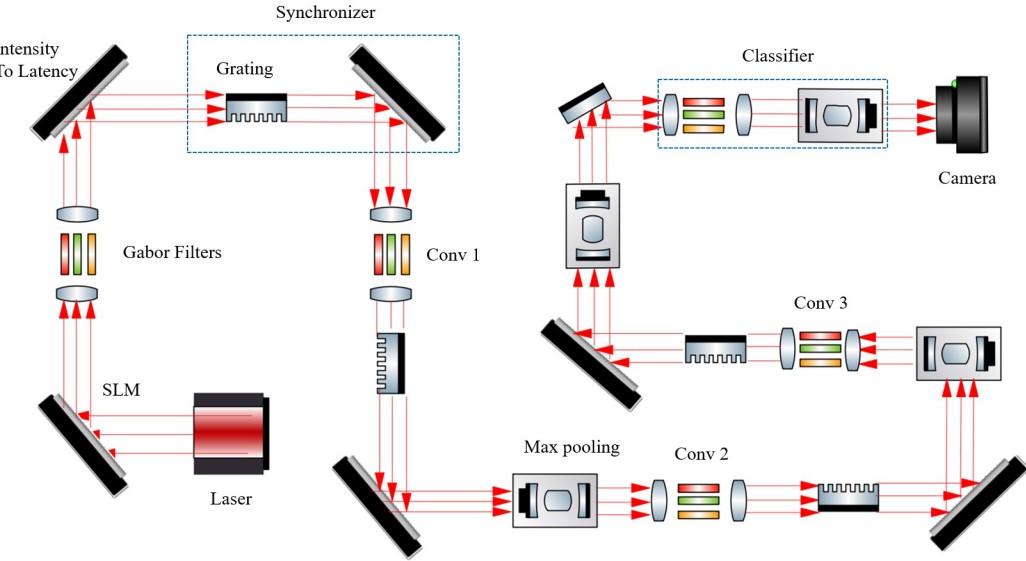

**Fig 2. The simulation of OSCNN involves loading data from a laser and SLM and simulating different optical free space components.**

the initial image. The image is loaded onto the SLM using electric wires. Next, we implement the Gabor filters using a 4f system as previously described. Following this, we employ an optical synchronizer, which utilizes a combination of a diffraction grating and an SLM. Subsequently, our network comprises three layers: a convolutional layer implemented using a 4f system, a synchronizer with a grating and SLMs, and a pooling function realized through a saturable absorber. Finally, we have a simple classifier consisting of linear and nonlinear functions implemented using a 4f system and a saturable absorber.

## 3 Simulation study

This section delves into the comprehensive simulations conducted to evaluate the OSCNN performance. The study extends to comparative analyses with other models, encompassing electrical and optical domains, free-space, and integrated approaches. A primary focus of the investigation lies in the in-depth analysis of the first-layer feature extractor kernels in both fixed and trainable configurations. Incorporating Gabor filters as convolutional kernels in the feature extractors of CNNs is particularly emphasized, owing to its biological inspiration from the human brain and inherent properties. This is further juxtaposed with comparisons involving other well-recognized filters such as Sobel, Canny, and Laplacian. Furthermore, the impact of noise on the input image is methodically examined, and the temporal consumption of electrical or optical resources by the OSCNN is quantitatively measured.

### 3.1 General properties of numerical simulation

The OSCNN model was subjected to rigorous training using the MNIST dataset, facilitated by a V100 Tesla GPU on the Google Colab platform. In this implementation, inspired by [25–28], the optical function of each of the structures is implemented, which means that, unlike the implementation of the wave-based in integrated ONNs, only the optical functions of which structures are implemented because the full wave analysis is practically impossible with standard simulators like COMSOL and Lumerical.

In the simulation environment, all the tools are placed one after the other, as shown in Fig 2, and the data during training is entered on the first SLM in sequence, just like an NN trained in an electronic computer. After measuring the output value, only the kernel values in all SLMs are updated, and practically, this is how BP is implemented and simulated instead of STDP. In this simulation, the convolution kernel sizes, which are the same as the SLM kernels, are 3,4 and 5, respectively, for three Optical SCNN layers. The desired phase shift in which of the SLMs in kernels is also related to the values of pixels and input data.

The cumulative duration for processing and training with the MNIST dataset amounted to 2 hours and 37 minutes, a timeframe that is markedly consistent with processing times associated with other electrical and optical models. The training process was executed through back-propagation, functionally equivalent to the Spike-Timing-Dependent Plasticity (STDP) process described in [24].

In addition to the MNIST dataset, the model was subjected to rigorous training and testing on the ETH-80 and Caltech datasets, both well-recognized benchmarks within the Spiking Neural Network (SNN) domain. Each optical module is rigorously formulated mathematically within these simulations and is referred to as a behavioral model.

## 3.2 Various feature extractors

The initial layer of the OSCNN is characterized by an array of Gabor filters, each possessing distinct spatial orientations and thicknesses. We benchmark the OSCNN against other Optical Neural Networks (ONNs) to provide a comprehensive comparative analysis of the feature extraction process. For instance, the Diffractive Deep Neural Network (D2NN) [29] leverages light diffraction properties employing apertures designed through the Huygens principle. In the OSCNN, Gabor filters are deployed to meticulously extract the most salient features embedded within the images. To this end, various feature extraction approaches are examined, including fixed Gabor filters devoid of training, trainable Gabor filters, and established filters like Canny, Laplacian, and Sobel. The MNIST, Caltech, and ETH-80 datasets employ these diverse feature extractors. The ensuing influence of these filters on the output accuracy is meticulously assessed and is presented in Table 1.

As depicted in Table 1, the outcomes reveal that the trainable Gabor filter, endowed with adaptable parameters about its filter length and central frequency, attains the highest performance among the filters tested. However, it is noteworthy that even the fixed Gabor filter consistently outperforms the alternative filters. This substantiates our assertion that Gabor filters can be regarded as reliable and effective feature extractors across various Optical Neural Networks (ONNs) and can be effectively deployed in the first layer as Convolutional Neural Network (CNN) kernels. For a further representation, Fig 3 showcases the output images generated by applying the Gabor input filter to an image from the dataset, exemplified by image number 8.

**Table 1. OSCNN accuracy with different feature extractors.**

| Filter | MNIST | Caltech | ETH-80 |
|---|---|---|---|
| Fixed Gabor | 91.4 | 88.3 | 84.7 |
| Trainable Gabor | **95.2** | **91.3** | **89.7** |
| Canny | 89.2 | 86.7 | 84.2 |
| Laplacian | 90.4 | 84.3 | 83.5 |
| Sobel | 86.7 | 82.1 | 79.4 |

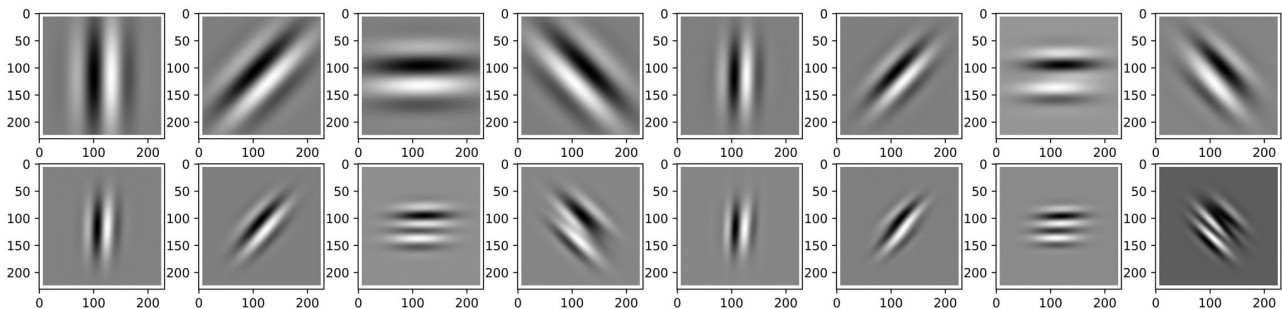

**Fig 3. Gabor filters are created as sine wave filters in four different directions, namely 0, π/2, π, and 3π/4 degrees.** These filters have different thicknesses and function as edge detectors, resembling the simple cells found in the primary visual cortex [20].

This comprehensive analysis underscores the exceptional efficacy of Gabor filters as optimal models for feature extraction in the initial layer of Optical Neural Networks (ONNs). Consequently, Gabor filters stand as a highly recommended choice for processing diverse data types, extending their applicability to various domains, including image analysis and the handling of temporal signals, such as audio, as expounded in [23].

### 3.3 Model performance

The evaluation of OSCNN's proficiency in object detection tasks encompassed its rigorous training and testing on three distinct datasets: MNIST, ETH 80, and Caltech. Subsequently, the results were meticulously juxtaposed against established electronic and optical models in the free-space and integrated domains. These comparative results are illustrated in Fig 4 and tabulated in Table 2.

Fig 4 offers a comparative overview, contrasting the performance of electronic implementations, specifically those of Alexnet and electrical models, with their optical free-space (OFS) counterparts. Notably, SNNs, whether instantiated electronically or optically, consistently manifest superior object detection performance when compared to CNNs.

Table 2 serves as a comprehensive comparative analysis, shedding light on the performance of various electrical and optical models tested on the MNIST dataset. The findings unequivocally underscore that OSCNN exhibits commendable performance levels in the context of object detection tasks. Nevertheless, it is crucial to acknowledge a notable disparity between the implementations of NNs and ONNs, a distinction prominently illustrated in Fig 4 and

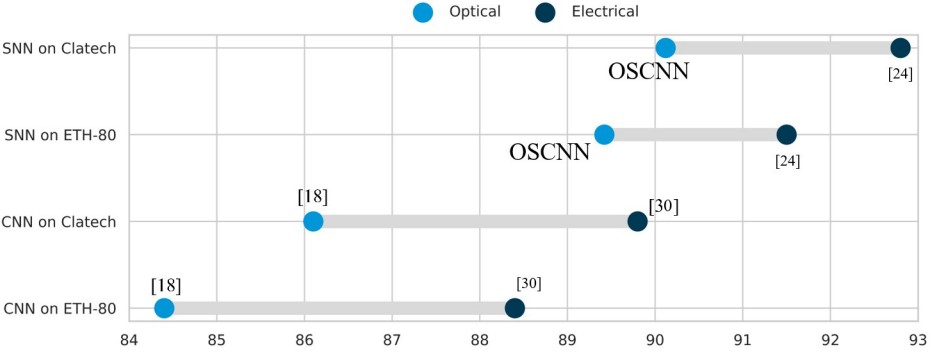

**Fig 4. Accuracy of different electrical and optical neural network on Caltech and ETH-80 data sets.**

**Table 2. Different implementations on MNIST object detection accuracy (%) with input image size.**

| Implementation | Approaches | Model | MNIST | Input size |
|---|---|---|---|---|
| Electrical | Perception | MLP [30] | 92.3 | $28 \times 28$ |
| | Convolution | AlexNet [31] | 98.3 | $28 \times 28$ |
| | Spiking | Kheradpisheh. et al [20] | 97.2 | $28 \times 28$ |
| Optical | Perception | Shen. et al [10] | 98.4 | $3 \times 3$ |
| | | D2NN [29] | 99.1 | $3 \times 3$ |
| | | Ryou. et al [25] | 86.7 | $3 \times 3$ |
| | Convolution | Bagherian. et al [9] | 87.7 | $3 \times 3$ |
| | | Sadeghzadeh. et al [26] | 97.6 | $28 \times 28$ |
| | Spiking | Feldmann. et al [18] | 98.3 | $3 \times 3$ |
| | | Xiang. et al [17] | 89.9 | $3 \times 3$ |
| | | OSCNN | 95.2 | $28 \times 28$ |

elaborated upon in Table 2. In this context, it is essential to highlight that FSO implementations consistently demonstrate accelerated processing speeds when juxtaposed against their integrated counterparts. Furthermore, FSO implementations exhibit remarkable data-handling capabilities, extending to vast datasets, including biological data, a feat that remains presently unattainable for integrated ONNs. Our model harnesses the inherent superiority of spiking neural networks, capitalizing on their event-driven nature to achieve unparalleled efficiency and accuracy in information processing. In addition, a distinctive feature of OSCNN lies in its capability to process large, real-data-sized images in free space. This showcases our model's scalability and positions it as a robust solution for applications dealing with extensive datasets. Unlike conventional ONNs, spiking neural networks efficiently process substantial real-world datasets. This makes it a compelling choice for applications demanding the analysis of large-scale images in free space. The spiking nature of OSCNN enables parallel information processing, making it particularly adept at handling the complexities of big real-data-sized images. This similar architecture ensures swift and efficient computations.

Our emphasis on processing big, real-data-sized images reflects a commitment to relevance in real-world applications. The optical SNN's proficiency in handling substantial datasets positions it as a valuable asset for tasks requiring real-world data analysis. The scalability and adaptability of our optical SNN set it apart, enabling seamless integration into systems that demand the processing of extensive real-data-sized images. This adaptability makes our model well-suited for a wide range of practical applications.

Recent studies have demonstrated impressive capabilities of free-space optical systems in neural network implementations. Two notable examples are the works of Ryou et al. [25] and Pierangeli et al. [32]. Ryou et al. presented a free-space optical neural network based on thermal atomic nonlinearity, utilizing a vapor cell as a nonlinear activation function, achieving high parallelism and energy efficiency in a multilayer perceptron architecture. Pierangeli et al. implemented a photonic extreme learning machine using free-space optical propagation, showcasing the potential for high-speed, large-scale neural network operations in optics.

While these works share the fundamental concept of leveraging free-space optics for neural computations with our OSCNN, our approach introduces several key innovations. Unlike the continuous-valued approaches in the aforementioned studies, our OSCNN incorporates spiking neural dynamics, introducing a temporal dimension to information processing that potentially offers enhanced energy efficiency and closer mimicry of biological neural systems. Additionally, our work specifically implements a convolutional neural network architecture in

the optical domain, which is particularly suited for image processing tasks and differs from the fully-connected architectures or extreme learning machines in the compared works.

We also introduce a unique synchronization technique using gratings and spatial light modulators. This addresses a key challenge in implementing timing-based neural networks in optical systems, which is not a concern in non-spiking architectures. Furthermore, our use of Gabor filters for the initial layer leverages the natural affinity between optical systems and Gabor-like transformations, potentially offering more efficient and effective feature extraction compared to traditional convolutional layers in both optical and electronic implementations.

The main contribution of our work lies not just in the use of a free-space optical system, but in the novel integration of spiking dynamics, convolutional architecture, and specialized optical components within this free-space framework. While the works by Ryou et al. and Pierangeli et al. demonstrate the power and flexibility of free-space optical systems for neural computations, our OSCNN extends these concepts into the realm of spiking neural networks and convolutional architectures.

This combination of features in our OSCNN potentially offers advantages in terms of energy efficiency, temporal information processing, and scalability, particularly for complex image processing tasks. Moreover, our approach opens up new avenues for exploring biologically-inspired computational paradigms in the optical domain, bridging the gap between traditional artificial neural networks and more brain-like computing systems.

As the field of optical neural networks continues to evolve, our work contributes to the growing body of knowledge on how to effectively leverage the unique properties of light for neural computation. Future research directions may include exploring hybrid systems that combine the strengths of different optical neural network approaches, further optimizing the energy efficiency and speed of optical spiking neural networks, and investigating more complex neuron models and learning algorithms in the optical domain.

## 3.4 Noise robustness

In this section, we delve into assessing the Optical Spiking Convolutional Neural Network (OSCNN) in terms of its resilience against input image noise. While our initial evaluation introduced white noise to the input images, representing various noise levels from 5% to 50%, the analysis encompasses a broader consideration of noise sources. We subjected OSCNN and several other electrical and optical Neural Networks (NNs) to evaluations, systematically varying the levels of input image noise. The results, depicted in Fig 5, provide a comprehensive understanding of OSCNN's performance under different noise conditions. Up to a noise level of 15%, OSCNN maintains a commendable accuracy, showcasing its ability to handle moderate noise levels effectively. However, beyond this threshold, accuracy experiences a rapid decline, ultimately converging to the chance level of 40To contextualize OSCNN's noise resilience, we conducted a comparative analysis with electrical CNN, electrical SNNs, and their optical counterparts. Fig 5 represents the relative robustness of the different networks under consideration. As shown in this Figure, OSCNN demonstrates superior robustness compared to electrical SNNs and optical SNNs. However, it is essential to acknowledge a limitation in robustness inherent to SNNs, both in the electrical and optical domains. As noise levels surpass 15%, the observed decline in accuracy prompts a critical reflection on the robustness challenges faced by SNNs.

This observation underscores the need for further research endeavors to enhance the noise resilience of SNNs, particularly in the context of optical implementations. While OSCNN exhibits competitive robustness, addressing the challenges associated with higher noise levels could potentially unlock new avenues for improvement. In conclusion, the noise resilience

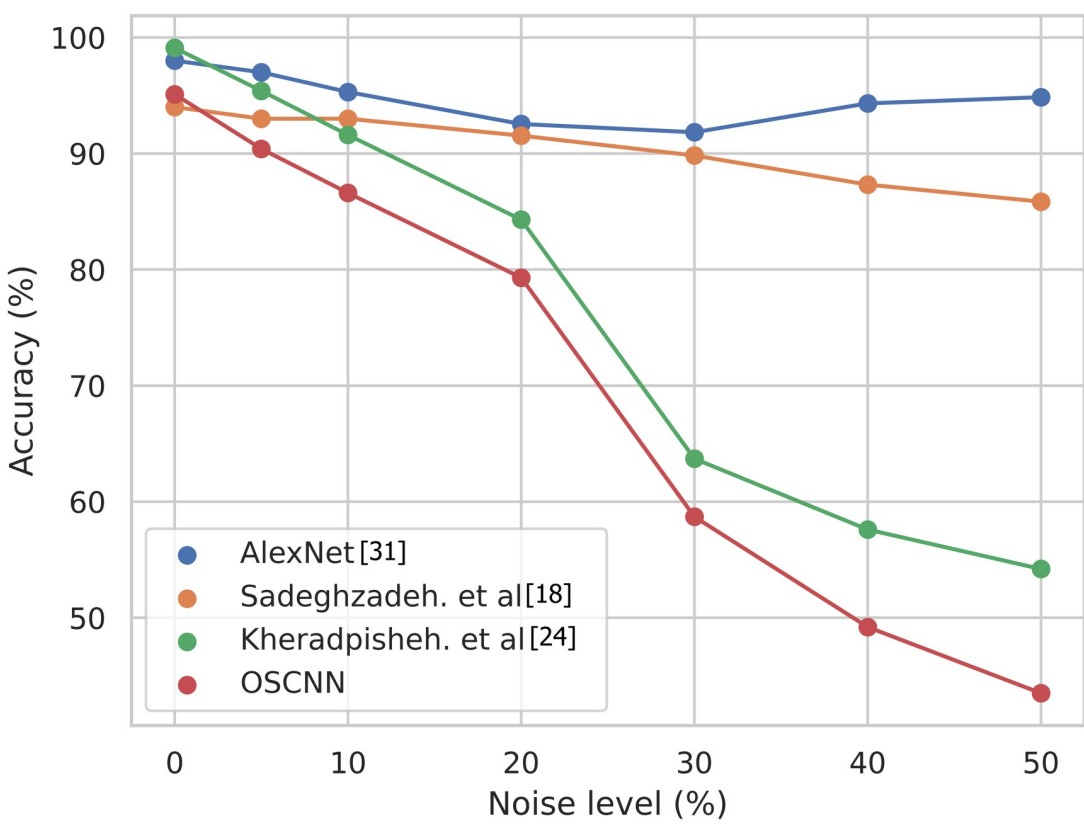

**Fig 5. Accuracy of different electrical and optical neural networks for various input noise levels.**

analysis extends beyond input image noise, highlighting OSCNN's strengths and areas for improvement. The comparative robustness analysis provides valuable insights, emphasizing the need for ongoing research to enhance the overall robustness of SNNs.

### 3.5 Speed analysis and comparative assessment

Optical computing presents a paradigm shift in processing capabilities, showcasing inherent advantages over its electrical counterparts, particularly in accelerated processing. To comprehensively evaluate the relative speed of the OSCNN, we conducted a detailed analysis of latency, deconstructing the estimation into various components.

$$
\begin{aligned}
t_{latency} &= t_{source} + t_{Gaborfilters} + t_{PM} + t_{Sync} \\
&+ t_{Conv1} + t_{Sync} + t_{Maxpooling} \\
&+ t_{Conv2} + t_{Sync} + t_{Maxpooling} \\
&+ t_{Conv3} + t_{Sync} + t_{Maxpooling} \\
&+ t_{classifier} + t_{camera} + t_{transferdata}
\end{aligned}
\tag{21}
$$

The latency estimation for OSCNN ($t_{latency}$) is meticulously deconstructed into several components, as outlined in Eq (9). Each component is carefully considered to provide a nuanced understanding of the processing speed. The modulation delay associated with input images

**Table 3. Comparison of optical and silicon-based neural network implementations.**

| Implementation | Architecture | Approx. Latency (28x28 image) |
|---|---|---|
| Optical | ANN [27] | 1.5 ms |
| | CNN [26] | 2.5 ms |
| | RNN [28] | 3.5 ms |
| | OSCNN | 2.44 ms |
| Silicon-based | ANN (CPU) | 3 ms |
| | ANN (GPU) [33] | 2 ms |
| | CNN (CPU) [33] | 15 ms |
| | CNN (GPU) [33] | 3 ms |
| | RNN (CPU) [33] | 30 ms |
| | RNN (GPU) [33] | 6 ms |
| | SNN [33] | 2 ms |

$(t_{source})$ is estimated at $1ms$, considering Spatial Light Modulators (SLMs) featuring a $1kHz$ switching frequency [26–28]. The processing time for the convolutional layer employing Gabor kernels (tGaborfilters) is estimated at approximately $5ps$, a negligible interval for practical considerations. The operation of converting intensity into latency $(t_{PM})$ is executed with an approximate duration of $1ms$ [26–28]. The synchronization system, comprising a grating layer and an SLM, incurs a cumulative processing time of approximately $2ms$, considering the dissonant layer's speed [26–28]. Optical propagation delays within the convolution and pooling layers $(t_{Conv1}$ and $t_{Maxpooling})$ are negligible, each estimated at approximately $5ps$ for 4f optical correlators [26–28]. The classification module introduces a minimal delay, with the nonlinear Saturable Absorber (SA) unit contributing approximately $25ns$ and the classification component exhibiting a delay of roughly $25ms$ [26–28]. The time required for high-speed commercial cameras to capture and convert output images into electrical data $(t_{camera})$ is estimated at $0.4ms$ for a camera capturing images at a rate of 2500 frames per second [26–28]. The communication interface introduces delays when transmitting camera output data to a computer $(t_{transferdata})$, estimated at $0.04ms$ for USB 3.1 Gen2 processing a 50 kB image.

The cumulative delay attributed to OSCNN is approximately $2.44ms$. This efficient processing time underscores the remarkable speed advantages of our optical spiking neural network. To further emphasize the superiority of OSCNN's speed, we compare this latency with the processing times typically observed in digital neural networks. The results reveal that OSCNN outperforms traditional digital systems, showcasing its exceptional speed in real-world applications. The detailed latency analysis and comparative evaluation demonstrate OSCNN's superior speed capabilities, positioning it as a promising advancement in optical computing. Table 3 compares optical and silicon networks, showing OSCNN's 2.44 ms latency outperforms CPUs and rivals GPUs.

## 3.6 Power consumption analysis

The power dynamics within the Optical Spiking Convolutional Neural Network (OSCNN) architecture have been explored in-depth, with insights drawn from prior research [26–28]. This investigation reveals that convolution, nonlinearity, and pooling operations contribute minimally to energy consumption, with the principal source lying in the domain of signal transmission [26–28]. To contextualize these findings, it is crucial to consider the power

demands associated with each pixel's capture, as elucidated in [26]:

$$P_{optical} = \frac{n^2 \times n_{kernel}}{\eta \times t^p} \tag{22}$$

The unit of power is expressed in micro-watts, where $n^2$ denotes the total number of pixels per 4f correlator system, $p$ signifies the number of optical elements traversing the optical path, $n_{kernel}$ is the arbitrary value indicating the number of kernels utilized by the convolutional layer, $t$ represents the fraction of incident power received by each optical element, and $\eta$ embodies the source efficiency.

However, a comprehensive evaluation of power consumption must extend beyond optical elements to consider the power requirements of high-speed cameras. Let $P_{camera}$ represents the power consumption of high-speed cameras capturing the optical output. For instance, if we assume $P_{camera} = 1.5 watts$ [26, 27], the total power consumption is now calculated as:

$$P_{total} = P_{optical} + P_{camera} \tag{23}$$

Conversely, when endeavoring to calculate the electric power consumption, careful consideration must be extended to the following factors, as expounded in [26]:

$$P_{electrical} = \beta \times n^2 \times k^2 \times n_{kernel} \times P_{switch} \tag{24}$$

In the Equation presented, the variable $\beta$ is determined by the architectural characteristics of the program, with $k$ signifying the kernel size and $P_{switch}$ representing the energy consumed by each operation. It's imperative to acknowledge that electronic components invariably escalate their power consumption as the kernel size expands. Conversely, the optical implementation of convolutional layers showcases a distinct trend: as the kernel size diminishes, the power consumption concurrently decreases. This divergence in behavior highlights a remarkable aspect of optical implementations—namely, their capacity for significant power savings, particularly in scenarios featuring large kernel sizes, in contrast to their electrical counterparts.

So the final value for the power consumption of OSCNN can be represented as:

$$P_{total} = P_{optical} + P_{camera} = \frac{28^2 \times 32}{0.5 \times 0.9^4} \times 10^{-6} + 1.5 \approx 1.587 \text{ W} \tag{25}$$

Also, a comparison between different open space optical neural network models and their equivalents implemented on the silicon system in test mode for a $28 \times 28$ image is presented in Table 4.

**Table 4. Comparison of power consumption in optical and silicon-based neural network implementations.**

| Implementation | Architecture | Power Consumption (W) |
|---|---|---|
| Free Space Optical NN | ANN [27] | 1 |
| | CNN [26] | 2 |
| | RNN [28] | 3 |
| | OSCNN | 1.6 |
| Silicon-based NN | ANN (CPU) | 30 |
| | ANN [34] (GPU) | 150 |
| | CNN (CPU) [4] | 100 |
| | CNN (GPU) [34] | 250 |
| | RNN (CPU) [34] | 100 |
| | RNN (GPU) [34] | 150 |
| | SNN [34] | 5 |

## 4 Conclusion and future works

This article marks a significant milestone by introducing the pioneering Optical Deep Spiking Convolutional Neural Network (OSCNN) model operating in free space. Drawing inspiration from the computational model of the human eye, this model excels in detecting patterns with commendable accuracy, processing speed, and power efficiency. An essential revelation from this study is the suggestion that, in the realm of optical neural networks tasked with image processing, the initial layer can be effectively realized by deploying Gabor filters, thereby revolutionizing the approach to feature extraction.

Furthermore, the remaining optical free-space components, encompassing the Intensity-to-Delay conversion and the synchronizer, were meticulously designed using readily available optical components. It is important to note that this optical model refrains from delving into the intricacies of biological neuron modeling and Spike-Timing-Dependent Plasticity (STDP) training. These aspects are deliberately deferred to future research endeavors, which will focus on designing a dedicated structure based on resonators or topological photonics. The objective is to authentically simulate the precise neuron model and subsequently introduce an array structure to usher in novel designs for convolutional layers, akin to the advancements witnessed in metasurface technology.

Our OSCNN model demonstrates significant advantages in terms of power consumption and processing speed. As shown in our comparative analysis, the OSCNN consumes only 1.6 W of power while achieving a processing speed of 2.44 ms, outperforming conventional electronic CNN implementations on GPUs, which typically consume 150-300 W with processing speeds of 1-5 ms. It also competes favorably with other free-space optical neural networks, which generally consume 1-5 W with processing speeds of 1-3 ms.

However, the development and implementation of such advanced optical neural networks present several challenges that warrant further investigation. The performance of free-space optical systems heavily relies on the precise alignment of optical components. To address this, we propose the use of advanced micro-positioning systems with nanometer-scale precision for initial alignment. Additionally, the implementation of active feedback control systems is crucial to maintain alignment during operation, compensating for thermal expansion and mechanical vibrations. Furthermore, the exploration of integrated optical designs that reduce the number of discrete components requiring alignment can significantly improve the robustness and reliability of the system, ensuring consistent performance while minimizing the need for frequent manual adjustments.

Optical signals are susceptible to various sources of noise and degradation. To mitigate these issues, we utilize high-quality optical components with anti-reflection coatings to minimize signal loss and stray reflections. The implementation of error correction algorithms and redundancy in data encoding improves signal integrity. Moreover, the incorporation of adaptive optics techniques allows for real-time compensation of wavefront distortions, and the development of noise-resistant coding schemes enhances the reliability of optical data transmission. These strategies significantly enhance the signal-to-noise ratio and overall system reliability.

The interface between optical and electronic domains is a critical aspect of our OSCNN design. To optimize this integration, we employ high-speed opto-electronic converters with minimal latency at critical junctions. Custom integrated circuits are designed for efficient processing of optically-derived signals, and advanced packaging techniques are utilized to minimize parasitic capacitances and signal degradation at optical-electronic interfaces. Additionally, the implementation of parallel processing architectures fully leverages the high bandwidth of optical signals, ensuring seamless integration while maintaining the speed and efficiency advantages of optical processing.

While laser sources introduce additional power and area requirements, our analysis shows that these overheads are outweighed by the overall system benefits. The utilization of highly efficient, compact semiconductor laser diodes minimizes both power consumption and footprint. Time-division multiplexing techniques are implemented to share laser sources across multiple channels, reducing the total number of required sources. Novel cooling strategies are developed to manage thermal loads efficiently, further reducing power overhead. A comparative analysis shows that despite these overheads, our OSCNN maintains a significant advantage in terms of overall power efficiency and processing speed.

The study relies on a well-established behavioral model instead of a numerical analysis of electromagnetic fields. The introduction of optical free-space deep-spiking convolutional neural network models takes a significant stride toward realizing high-powered, high-speed processors inspired by the human brain. This endeavor propels us closer to developing an artificial brain, manifesting itself in an optical form endowed with formidable computational capabilities.

In conclusion, while the challenges in implementing optical neural networks are substantial, our OSCNN design demonstrates that these can be effectively addressed. The resulting system offers a compelling combination of power efficiency and processing speed, positioning it as a promising approach for next-generation neural network implementations. Future research will focus on overcoming these hurdles, exploring more complex neuron models, advanced training techniques, and novel optical architectures to further push the boundaries of neural network performance and efficiency.

## Author Contributions

**Conceptualization:** Reyhane Ahmadi, Somayyeh Koohi.

**Data curation:** Reyhane Ahmadi, Amirreza Ahmadnejad.

**Investigation:** Reyhane Ahmadi, Amirreza Ahmadnejad.

**Methodology:** Amirreza Ahmadnejad.

**Project administration:** Amirreza Ahmadnejad.

**Software:** Reyhane Ahmadi.

**Supervision:** Somayyeh Koohi.

**Validation:** Reyhane Ahmadi, Amirreza Ahmadnejad.

**Visualization:** Amirreza Ahmadnejad.

**Writing – original draft:** Amirreza Ahmadnejad.

**Writing – review & editing:** Amirreza Ahmadnejad.

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
