## [Decision Letter · Decision Letter 0]

4 Jul 2024

PONE-D-24-01153Free-Space Optical Spiking Neural NetworkPLOS ONE

Dear Dr. Koohi,

Thank you for submitting your manuscript to PLOS ONE. After careful consideration, we feel that it has merit but does not fully meet PLOS ONE’s publication criteria as it currently stands. Therefore, we invite you to submit a revised version of the manuscript that addresses the points raised during the review process.

We look forward to receiving your revised manuscript.

Kind regards,

Farooq Ahmad Khanday, Ph.D.

Academic Editor

PLOS ONE

Reviewers' comments:

Reviewer's Responses to Questions

**Comments to the Author**

1. Is the manuscript technically sound, and do the data support the conclusions?

Reviewer #1: Partly

Reviewer #2: Partly

2. Has the statistical analysis been performed appropriately and rigorously? 

Reviewer #1: N/A

Reviewer #2: Yes

3. Have the authors made all data underlying the findings in their manuscript fully available?

Reviewer #1: No

Reviewer #2: Yes

4. Is the manuscript presented in an intelligible fashion and written in standard English?

Reviewer #1: Yes

Reviewer #2: Yes

5. Review Comments to the Author

Reviewer #1: 1. Please discuss more in detail what are the limitations of the related work and how these issues are addressed in the proposed method.

2. Please discuss clearly what are the novel contributions in this paper.

3. Please discuss the proposed method more in detail through a detailed top-level algorithm describing all the operations involved.

4. Please describe in more detail the experimental setup and tool flow used to conduct the experiments. If possible, please open-source the code to allow reproducibility of the experiments.

5. In the result section, each figure and table should have lists of observation points to discuss the observations derived from the results.

6. Please discuss more in detail the potential impact at large scale and future works derived from this paper.

Reviewer #2: The authors have introduced an ultra-fast and energy-efficient free-space Optical deep Spiking Convolutional Neural Network, highlighting advancements in computational efficiency, biological plausibility, temporal encoding, hardware scalability, low-power embedded applications, and free-space simulation. I have the following comments:

1. The main contribution of this work is the use of a free-space optical system for neural networks, independent of the neural network type. How does this basic idea differ from existing simulation and experimental-based studies, such as the following papers:

   - Albert Ryou, James Whitehead, Maksym Zhelyeznyakov, Paul Anderson, Cem Keskin, Michal Bajcsy, and Arka Majumdar, "Free-space optical neural network based on thermal atomic nonlinearity," Photon. Res. 9, B128-B134 (2021)

   - Davide Pierangeli, Giulia Marcucci, and Claudio Conti, "Photonic extreme learning machine by free-space optical propagation," Photon. Res. 9, 1446-1454 (2021)

2. The proposed study is simulation-based, making it difficult to accurately estimate the actual implementation feasibility and performance.

3. The authors should provide a more detailed explanation of how (i) intensity to delay conversion, (ii) Gabor filter implementation, (iii) neuron and weight storage elements, and (iv) convolution can be implemented in hardware.

4. While the results are well presented, the authors need to integrate the latency and power consumption results of the proposed approach with those of other conventional state-of-the-art methods. The difference between conventional and optical convolutional computation should be highlighted in the comparative analysis.

5. The authors should discuss in detail how the following challenges can be addressed: (i) precise alignment of components (ii) noise and degradation of optical signals (iii) combining optical and electronic components, which will be required at almost every stage(iv) Whats the actual power and area overhead of laser sources as compared to the conventional techniques? These overheads can diminish the proposed advantages.

6. PLOS authors have the option to publish the peer review history of their article (what does this mean?). If published, this will include your full peer review and any attached files.

Reviewer #1: No

Reviewer #2: No

---

## [Author Response · Author response to Decision Letter 0]

6 Sep 2024

List of Changes

Abstract

We have made revisions to the Abstract, outlined below:

Neuromorphic engineering has emerged as a promising avenue for developing brain-inspired computational systems. However, conventional electronic AI-based processors often encounter challenges related to processing speed and thermal dissipation. As an alternative, optical implementations of such processors have been proposed, capitalizing on the intrinsic information-processing capabilities of light. Among the various Optical Neural Networks (ONNs) explored within the realm of optical neuromorphic engineering, Spiking Neural Networks (SNNs) have exhibited notable success in emulating the computational principles of the human brain. The event-based spiking nature of optical SNNs offers capabilities in low-power operation, speed, temporal processing, analog computing, and hardware efficiency that are difficult or impossible to match with other ONN types.

In this work, we introduce the pioneering Free-space Optical Deep Spiking Convolutional Neural Network (OSCNN), a novel approach inspired by the computational model of the human eye. Our OSCNN leverages free-space optics to enhance power efficiency and processing speed while maintaining high accuracy in pattern detection. Specifically, our model employs Gabor filters in the initial layer for effective feature extraction, and utilizes optical components such as Intensity-to-Delay conversion and a synchronizer, designed using readily available optical components. The OSCNN was rigorously tested on benchmark datasets, including MNIST, ETH80, and Caltech, demonstrating competitive classification accuracy.

This part is added to emphasize achievements and highlights, as well as add speed and power consumption numbers.

Our comparative analysis reveals that the OSCNN consumes only 1.6 W of power with a processing speed of 2.44 ms, significantly outperforming conventional electronic CNNs on GPUs, which typically consume 150-300 W with processing speeds of 1-5 ms, and competing favorably with other free-space ONNs.

Our contributions include addressing several key challenges in optical neural network implementation. To ensure nanometer-scale precision in component alignment, we propose advanced micro-positioning systems and active feedback control mechanisms. To enhance signal integrity, we employ high-quality optical components, error correction algorithms, adaptive optics, and noise-resistant coding schemes. The integration of optical and electronic components is optimized through the design of high-speed opto-electronic converters, custom integrated circuits, and advanced packaging techniques. Moreover, we utilize highly efficient, compact semiconductor laser diodes and develop novel cooling strategies to minimize power consumption and footprint.

Introduction

We have revised the Introduction section to more effectively address the need for Spiking Neural Networks (SNNs) and Optical Free Space Optical Neural Networks (ONNs). The updated introduction now emphasizes the limitations of traditional electronic and integrated photonic systems, particularly in terms of processing speed, power consumption, and scalability.

We highlight why SNNs are crucial by detailing their ability to better mimic the brain’s event-driven processing and robustness to noisy inputs. Additionally, we address why Optical Free Space (OFS) ONNs are needed, focusing on their advantages in scalability, reconfigurability, and reduced optical loss, which are essential for handling large-scale, high-speed data processing tasks. By expanding on these points, the revised section underscores the importance of our proposed approach and its potential impact in overcoming the limitations of existing technologies. Here is the new revised introduction:}

The human brain represents a profoundly intricate and remarkable biological entity. The endeavor to engineer a computational processor possessing commensurate attributes in power, precision, integration, and speed has perennially constituted a paramount aspiration for processor designers. Neuromorphic Engineering stands as a foundational paradigm facilitating the realization of such processors, primarily through the incorporation of neural network architectures [1-3]. Despite the notable achievements resulting from this approach [4-7], the central challenge in processor design endures as the demand for processing voluminous datasets continues to burgeon. One of the fundamental problems that has led researchers to the optical implementation of processing and even neuromorphic structures in recent years is related to Moore's law [7] .

To address this persistent challenge, Optical Neuromorphic Engineering has emerged as a novel and innovative domain [8]. Optical Neuromorphic Engineering exploits light's distinctive attributes, including its exceptional propagation speed and the extended degrees of freedom it affords in comparison to electrons, encompassing characteristics such as frequency, phase, polarization, and mode. Furthermore, optical systems manifest reduced loss, rendering them remarkably compelling for the design and construction of Optical Neural Networks (ONNs). The most significant advancements in recent years are documented in [9-12] and comprehensively reviewed in [13-15].

Various Optical Neural Networks (ONNs) have been explored within the realm of optical neuromorphic engineering. Among these, Spiking Neural Networks (SNNs) have exhibited notable success in emulating the computational principles of the human brain. SNNs, which constitute a class of neural networks that emulate the structural and functional aspects of the human brain, have garnered significant attention in this context. Numerous optical models have been proposed for implementing SNNs; however, these models have been primarily integrated. Several intricate designs featuring components such as Vertical-Cavity Surface-Emitting Lasers (VCSELs) [21], micro-ring resonators [22], and phase-change materials [23] have been suggested. Nonetheless, these designs prove ill-suited for the demanding task of high-volume data processing, thereby underscoring the formidable challenges encountered within Optical Neuromorphic Engineering.

The potential need for optical-spiking neural networks arises from the unique combination of optical and spiking neural network properties, offering specific advantages in certain applications. While all ONNs share standard optical features, incorporating spiking neural network characteristics introduces additional benefits, which can mention as help to have Event-Driven Processor Architectures, Temporal Information Capture, and Enhanced Robustness to Noisy Inputs.

This part is added to emphasize main reasons for our interest in implementing SNN is expressed as free space optical NN

Implementing Spiking Neural Networks (SNNs) in an Optical Free Space format (OFS) format offers several advantages compared to integrated photonic implementations:

Increased Scalability: OFS systems can accommodate vast numbers of optical components, enabling the realization of networks with millions or billions of spiking neurons. This massive scalability is challenging for integrated photonics.

Flexible Reconfigurability: The OFS approach simplifies reconfiguring and modifying network connectivity patterns by adjusting optical components. Integrated systems are more rigid once fabricated. Additionally, the broader operating wavelengths of OFS implementations can leverage a more comprehensive range of optical wavelengths, from UV to far infrared, whereas on-chip photonics have much narrower wavelength compatibility.

 Simplified Training: Specific OFS architectures allow direct intensity-to-spike training via simple optical nonlinearities, whereas integrated training often requires more complex components.

Lower Optical Loss: OFS systems can reduce loss, allowing for more extensive networks. Moreover, hybrid electronic-photonic operation interfaces between OFS optical components and traditional electronic neurons/synapses provide unique hybrid SNN capabilities.

It is imperative to establish precise mathematical models for each constituent component to develop an OFS model for SNNs. Remarkably successful models rooted in neuroscience [24,25] have been devised, emulating the structural attributes responsible for object detection within the human eye. In this paper, we aim to draw upon these well-established models as a source of inspiration for designing OFS components, thereby facilitating the simulation of the Free-Space Optical deep Spiking Convolutional Neural Network (OSCNN). To the best of our knowledge, OSCNN marks the inaugural foray into the realm of OFS modeling for SNNs, encompassing critical elements such as Gabor filters for feature extraction, intensity-to-delay conversion, synchronization mechanisms, convolution layers, max-pooling procedures, and a classification framework. A dedicated module was introduced to execute the intensity-to-delay conversion, employing a Spatial Light Modulator (SLM) after the feature extractor layer.

Moreover, an optical synchronizer has been meticulously devised to address the temporal processing aspects inherent to optic signals. Ensuring that the time order of signals remains intact post-convolution, this synchronizer draws inspiration from the Free-Space Optical delay line concept [26]. The performance of OSCNN was systematically evaluated across three distinct datasets: MNIST, Caltech, and ETH80. OSCNN demonstrated notable achievements, boasting significant performance metrics compared to electronic Neural Networks (NNs) and Optical Neural Networks (ONNs). Notably, Gabor filters were harnessed as feature extractors in the initial model layer, with evaluations conducted under both trained and fixed conditions. While alternative filters, such as Canny, Laplacian, and Sobel, were explored as feature extraction mechanisms, the most favorable outcomes for OSCNN were attained using Gabor filters. The results underscore the versatility of Gabor-form convolutional kernels, revealing their efficacy in image and time-series processing applications [27,27].

This part is added here for the order of the article which will be discussed in the next chapters

The general structure of this article is as follows: firstly, the optical model of the OSCNN is discussed. We have designed this model based on separate components, which have been explained in detail regarding the mathematics and how to model each component. Then, the numerical simulation results are presented, including a comparison of the overall performance of integrated optical neural networks and free space in the context of data classification, as well as an analysis of the effectiveness concerning noise, power, and required speed. Finally, we discuss future works and present our conclusions.

OSCNN Model

In our previous paper, we focused primarily on the mathematical models of each component within the Optical Free Space Spiking Convolutional Neural Network (OSCNN). In this revision, we have expanded our discussion to include detailed implementation strategies for each module, along with their mathematical properties and practical aspects. This enhancement provides a clearer understanding of how the theoretical models are translated into practical systems.

Due to equipment constraints, our current work relies on simulations to demonstrate the functionality of OSCNN. While this approach follows traditional and established methods in free space optical networks, we aim to elucidate the practical steps needed for actual implementation. This will serve as a guide for future work, whether by our team or other researchers, to realize this network in real-world applications.

Additionally, our work is inspired by the well-known Spiking Convolutional Neural Network (SCNN) mathematical model. We not only explain the mathematical foundations of this model but also detail how our OSCNN design draws from and adapts these principles. By integrating both the theoretical and practical aspects, we provide a comprehensive framework for implementing spiking neural network methodologies in optical free space environments. Here is the new revised section:

In delineating the architectural framework of the Optical Free Space Spiking Convolutional Neural Network (OSCNN), it is imperative to establish a comprehensive model for neurons within Spiking Neural Networks (SNNs, also known as integrate-and-fire models). SNNs fundamentally operate on spike-timing-dependent plasticity (STDP) principles, a paradigm necessitating optical modeling to faithfully replicate its mechanisms. OSCNN adopts a specialized variation of computational neurons, acknowledging that forthcoming research will delve into developing a more precise neuron model. As an alternative to STDP, backpropagation (BP) can be employed for training the OSCNN model, as described in [29]. Subsequently, this section elaborates on the mathematical underpinnings of each module within OSCNN, along with their optical equivalents. The holistic structure of the OSCNN model is illustrated in Figure 1 for a comprehensive overview.

This part is explained for a more detailed explanation of Figure 1

As shown in Figure 1, which is inspired by [25], the main features are first extracted from the input data. In the case of SNNs, these features are represented as a series of lines with different thicknesses, indicating their importance. These thickness-based features are then converted into fuzzy features, where those with higher value and importance are released into the model earlier.

A crucial aspect of this model is the synchronization of different phases or speeds, ensuring that all features begin propagating through the main neural network simultaneously. For this reason, a synchronizer is employed. 

The main OSCNN model comprises the convolution layer, synchronizer, and pooling layer. Finally, a classifier categorizes the input data.

One of the main problems that existed in the previous text was the failure to express the mathematics and inspired model for the optical implementation

In this part, the inspired model is explained.

This model, which is implemented on silicon systems, has been able to provide very acceptable results, and the main cornerstone of our optical implementation model is free space.

To further explain the main idea of [25] during test, the mathematical procedure can be expanded as follows.

First, they apply Difference of Gaussians (DoG) filters to the input image to detect contrasts. This can be mathematically represented as:

Next, the strength of these contrasts is encoded into spike times. The encoding is such that higher contrasts result in shorter latencies:

The learning mechanism employed is Spike-Timing-Dependent Plasticity (STDP), which updates the synaptic weights based on the timing of pre- and postsynaptic spikes:

A Winner-Take-All mechanism ensures that only the earliest spikes are considered, suppressing later spikes:

Pooling layers provide translation invariance and compress visual information. Translation invariance is achieved using maximum operations:}

The earliest spike times are propagated to reduce data complexity:

The classifier interprets the activity of neurons in the final pooling layer to determine the object category:

In the following, our goal is to simulate and present the optical equivalent of each of these functions, inspired by the mathematics and functions used to implement the [25]. The following sections provide a detailed explanation of each block and their free space optical equivalent design.

A new title was chosen for this part

Implementation of Gabor Filters Using an Optical 4f System

A Gabor filter can be implemented using an optical 4f system, which is a common setup for performing Fourier transforms and convolutions optically. The 4f system consists of two lenses with focal length f, placed at a distance 2f apart. This system utilizes the Fourier transform properties of lenses to perform the convolution of an image with a Gabor filter in the Fourier domain. The key components of the setup include the input image X(x, y) placed at the input plane. A lens with focal length f performs the Fourier transform of the input

---

## [Decision Letter · Decision Letter 1]

28 Oct 2024

Free-Space Optical Spiking Neural Network

PONE-D-24-01153R1

Dear Dr. Koohi,

We’re pleased to inform you that your manuscript has been judged scientifically suitable for publication and will be formally accepted for publication once it meets all outstanding technical requirements.

Kind regards,

Farooq Ahmad Khanday, Ph.D.

Academic Editor

PLOS ONE

Additional Editor Comments (optional):

Reviewers' comments:

Reviewer's Responses to Questions

**Comments to the Author**

1. If the authors have adequately addressed your comments raised in a previous round of review and you feel that this manuscript is now acceptable for publication, you may indicate that here to bypass the “Comments to the Author” section, enter your conflict of interest statement in the “Confidential to Editor” section, and submit your "Accept" recommendation.

Reviewer #1: All comments have been addressed

Reviewer #2: All comments have been addressed

2. Is the manuscript technically sound, and do the data support the conclusions?

Reviewer #1: (No Response)

Reviewer #2: Yes

3. Has the statistical analysis been performed appropriately and rigorously? 

Reviewer #1: (No Response)

Reviewer #2: N/A

4. Have the authors made all data underlying the findings in their manuscript fully available?

Reviewer #1: (No Response)

Reviewer #2: Yes

5. Is the manuscript presented in an intelligible fashion and written in standard English?

Reviewer #1: (No Response)

Reviewer #2: Yes

6. Review Comments to the Author

Reviewer #1: The authors have addressed the reviewers’ comments.

Reviewer #2: The authors have addressed my comments satisfactorily. I have no further comments. The manuscript can be accepted for publication.

7. PLOS authors have the option to publish the peer review history of their article (what does this mean?). If published, this will include your full peer review and any attached files.

Reviewer #1: No

Reviewer #2: No

---

## [Editor Report · Acceptance letter]

19 Nov 2024

PONE-D-24-01153R1 

PLOS ONE

Dear Dr. Koohi, 

I'm pleased to inform you that your manuscript has been deemed suitable for publication in PLOS ONE. Congratulations! Your manuscript is now being handed over to our production team.

Kind regards, 

on behalf of

Dr. Farooq Ahmad Khanday 

Academic Editor

PLOS ONE